# Care-related quality of life of informal caregivers of stroke survivors: Cross-sectional analysis of a randomized clinical trial

Lorena Villa-García[1,2,3], Mercè Salvat-Plana[4,5], John Slof[6], Natalia Pérez de la Ossa[4,7], Sònia Abilleira[4,5], Marc Ribó[8], Verónica Hidalgo-Benítez[9], Marco Inzitari[1,10], Aida Ribera[1,5]*

1 REFiT Bcn Research Group, Parc Sanitari Pere Virgili and Vall d'Hebron Institute of Research (VHIR), Barcelona, Spain, 2 Doctorate Program, Department of Medicine, Universitat Autònoma de Barcelona, Barcelona, Spain, 3 QIDA, Sabadell, Barcelona, Spain, 4 Stroke Programme, Catalan Health Department, Agency for Health Quality and Assessment of Catalonia, Barcelona, Spain, 5 CIBER de Epidemiología y Salud Pública (CIBERESP), Instituto de Salud Carlos III, Madrid, Spain, 6 Department of Business, Universitat Autònoma de Barcelona, Barcelona, Spain, 7 Department of Neurology, Stroke Unit, Hospital Universitari Germans Trias i Pujol, Badalona, Spain, 8 Stroke Unit, Department of Neurology, Hospital Vall d'Hebron, Universitat Autònoma de Barcelona, Barcelona, Spain, 9 Hospital Parc de Salut Mar, Barcelona, Spain, 10 Faculty of Health Sciences, Universitat Oberta de Catalunya (UOC), Barcelona, Spain

* ariberas@perevirgili.cat

## Abstract

### Purpose

We aimed to describe the intensity of care and its consequences on informal caregivers of stroke survivors according to the degree of care receivers' functional dependence for activities of daily living; and to identify the factors associated with caregivers' care-related quality of life.

### Methods

Cross-sectional analysis of prospective data collected in a cost-utility study alongside the RACECAT trial in Catalonia (Spain). One-hundred and thirty-two care receiver-caregiver pairs were interviewed six months after stroke. Functional dependence for activities of daily living was measured with the Barthel index. We assessed caregivers care-related quality of life with the CarerQoL, which measures seven dimensions of subjective burden (CarerQoL-7D) and a happiness score (CarerQoL-VAS). We evaluated the association between characteristics of informal caregivers, characteristics of care receivers, and intensity of care, and the caregiver's care-related quality of life (subjective burden and happiness) in a hypothesized model using a structural equation model.

### Results

Of the 132 caregivers, 74,2% were women with an average age of 59.4 ± 12.5 years. The 56.8% of them were spouses. The care intensity ranged from a mean of 24h/week for mild to 40h/week for severe dependence. Most caregivers (76.3%) were satisfied with their task, regardless of dependence, but showed increasing problems in caring for severely

ethical restrictions (such as the sensitive nature of some data). Restrictions were imposed by the Ethics Committee. The main points of contact for handling requests for access to the data in this manuscript are: the Vall d'hebron ethics committee (ceic@vhir), the corresponding author (Aida ribera: aribera@perevirgili.cat).

**Funding:** The study received funding by the Fundació Marató de TV3 (ref. 19/U/2017; authors AR who received. LV-G was funded by the Industrial Doctorates Program [reference 2020 DI 76], promoted by the Government of Catalonia, Spain. The funders had no role in study design, data collection and analysis, decision to publish, or preparation of the manuscript.

**Competing interests:** The authors have declared that no competing interests exist.

**Abbreviations:** BI, Barthel Index; CarerQoL, Care-related Quality of Life Instrument; CFI, comparative fit index; CG, Informal caregiver; CI, Confidence Interval; CD, coefficient of determination; NIHSS, National Institute of Health Stroke Scale; mRs, modified Rankin Scale; RC, care receivers; RMSEA, root mean square error of approximation; SRMR, Standardised Root Mean Square Residual.

dependent persons. Being a woman (coeff. -0.23; 95%CI: -0.40, -0.07), spending more time in care tasks (coeff -0.37; -0.53, -0.21) and care receiver need of constant supervision (coeff 0.31; -0.47, -0.14) were associated with higher burden of care, irrespective of the degree of dependence. Caregiver burden (coeff 0.46; 0.30–0.61) and care receiver anxiety or depression (coeff -0.19; -0.34, -0.03) were associated with lower caregiver happiness.

## Conclusions

The findings suggest the importance of developing mainly two types of support interventions for caregivers: respite and psychosocial support. Especially for women with high caring burden and/or caring for persons with high levels of anxiety or depression.

## Introduction

Stroke is a major cause of disability and death globally [1]. Stroke survivors often require long-term care, being family members or close relatives the main providers of care for stroke survivors living at home [2]. Thus informal care, also known as unpaid care or family care, is an increasingly relevant issue for health and social care policies.

Because of the sudden onset of stroke, informal caregivers (CGs) take on a role and responsibilities for which they are not always prepared [2, 3]. While providing care can be positive and rewarding [4], it has negative impacts on caregivers' health, well-being and quality of life [3, 5], generating the so-called *caregiver burden* [6].

CG burden is a multidimensional construct characterized by complex interactions of factors: the characteristics of CGs (gender, education, socio-economic and health status), care receivers (CR) (health status and the level of dependency), intensity and type of care, the relationship between the CG and the CR, and the potential barriers to health and social services [2, 3, 6, 7]. As a result of the complex interaction of these factors, CGs may experience impaired mental or physical health, and reduced quality of life [3, 5]. Furthermore, their participation in leisure activities, social relationships, and/or paid work may be limited due to caregiving responsibilities [3, 5, 8].

In addition to direct healthcare cost of stroke [9], the burden of informal care represents a considerable cost that has often been overlooked and undervalued [10, 11]. In recent years, there is growing literature focusing on factors associated with caregiver burden [2] and growing interest in incorporating informal care into economic evaluations [10, 12, 13]. However, less attention has been paid to the quality of life of the CGs [14] and its inclusion in economic evaluations has been limited [15, 16].

The WHO Health Organization defines QOL as an individual's perception of his or her own life in the context of the culture and value systems in which they live and in relation to their expectations, norms, and concerns [17]. Measures of QoL can be generic (i.e., designed to be used across disorders and health states) or disease/affliction-specific (i.e., related to a single disorder or health state). Several methods have been proposed to determine an appropriate definition and measurement of QoL in the context of informal care, however, as yet no unified agreement has been established. To overcome all these limitations, instruments have been developed to measure CG-related quality of life [18]. The CarerQol measures CG burden and CG subjective well-being (happiness) [19]. CarerQol allows for the analysis of the source of positive and negative impacts and assesses the impact of caregiving in economic terms. It may be useful in providing greater insight into the needs of CGs of stroke survivors, and in

incorporating CGs quality of life into economic evaluations [20]. However, to our knowledge there are no studies evaluating the CG-related quality of life of CG of patients with stoke sequelae in terms of burden and well-being as assessed by the CarerQoL. The future availability of informal care is of great social concern [21]. Given the current transformations of health and social care integration [22] and the increasing importance of the role of CGs, it is essential to learn the factors that influence wellbeing of CG. This can help to formulate person-centered, sustainable, and integrated care interventions.

The objectives of this study were: 1) to describe the intensity of care and its consequences on the CG according to the degree of dependence of persons surviving 6 months after a stroke; 2) to identify the factors associated with the care-related quality of life of CGs.

## Methods

### Study design

Cross-sectional descriptive study. This study was performed in the context of a cost-utility analysis parallel to the RACECAT (Transfer to the Closest Local Stroke Center vs Direct Transfer to Endovascular Stroke Center of Acute Stroke Patients with Suspected Large Vessel Occlusion in the Catalan Territory) trial (ClinicalTrials.gob NCT02795962). The RACECAT trial [23] compared direct transportation of patients with suspected large-vessel occlusion stroke to a thrombectomy-capable center with the usual initial transport to the nearest local referral center [23, 24]. There was no significant difference in 90-day neurological outcome between groups. In addition to clinical data, the necessary data to estimate direct and indirect costs and utilities was collected in a subsample of the RACECAT trial. All findings were reported in accordance with the STROBE guidelines (S1 Appendix).

### Study population

From the subsample included in the cost-utility analysis, we selected the dyads of CGs and CRs surviving 6 months after stroke. An informal CG was considered as a relative or close person to the CR who reported he/she was providing care to the CR, without any type of contract or remuneration.

We included patients 18 years old or more, with clinical suspicion of large-vessel occlusion stroke identified by the RACE scale score > 4 [25] in the prehospital setting, without a significant functional disability before stroke.

### Measures

We collected sociodemographic information of the CG and the CR: sex, age, educational level, marital status, relationship between CG and CR, cohabitation and household composition.

### Characteristics of informal caregivers

The main outcome was caregiving-related quality of life measured with the Spanish version of the CarerQoL [26]. The CarerQoL has two parts. The CarerQoL-7D [19, 20] measures subjective burden in 7 dimensions, 5 related to problems: relational, mental health, physical health, financial and combining daily activities with caring; and two positive dimensions: feeling of fulfillment and receiving support from others; with three response levels: "no", "a little" or "a lot". The CarerQoL Visual Analogic Scale (CarerQoL-VAS) measures well-being in terms of happiness ranging from 0 (completely unhappy) to 10 (completely happy) [19]. We used the Netherland's tariff [27] to transform the CarerQoL-7D profile into a utility weights from 0 (worst situation) to 100 (best situation).

## Characteristics and intensity of caregiving

We collected information on the total hours per week dedicated to caregiving, the type of care, the degree of supervision, and whether care was shared with other CG or professional caregiver or day care center.

## Characteristics of care receivers

We measured the degree of post-stroke dependence in activities of daily living with the Barthel index (BI). The BI score ranges from 0 (total dependence) to 100 (independent) [28]. We categorized BI into severe (<35), moderate (40–55) and mild (>60) dependence. The Health-related quality of life of patients was measured using the EuroQol-5D-5L, which measures 5 dimensions of quality of life (mobility, self-care, usual activities, pain/discomfort, and anxiety/depression) in 5 severity levels (1 = "no problems" to 5 = "extreme problems") [29]. It also includes a visual analogical scale (EQ-VAS) scored from 0 (the worst health you can imagine) to 100 (the best health you can imagine). The modified Rankin scale (mRS) was used for functional assessment after a cerebrovascular event. Scores range from 0 (no symptoms) to 6 (death) [30]. Stroke severity was assessed at diagnosis using the National Institute of Health Stroke Scale (NIHSS) [31].

## Data collection

We included patients randomized between September 2017 and January 2019 A single trained research nurse interviewed the CR and the CG separately in two telephone calls and at 3 and 6 months after randomization, following a standardized questionnaire adapted from de iVICQ [32]. Up to three attempts were made to contact participants. Prior to the start of the telephone interview, oral informed consent was obtained and recorded from all participants.

## Statistical analysis

We calculated frequencies for categorical variables and means and standard deviation and/or medians and interquartile ranges for continuous variables. To test for significance of differences between levels of dependence we used Chi squared test for categorical variables and ANOVA or Kruskal-Walls test for quantitative variables.

To identify the determinants of care-related quality of life we performed Structural Equation Models (SEM) in STATA 17.00. SEM is a statistical technique that combines path analysis, confirmatory factor analysis, and regression analysis to analyze the interrelationships of independent variables and their indirect effects through other variables. SEM allows investigators to examine a more complex and diverse network of variables, including both observed and latent ones. This method is suitable for analyzing the numerous factors that impact HRQOL, as it can simultaneously examine all of them. HRQOL is influenced by various factors, making SEM an appropriate method for analysis [33, 34].

A preliminary conceptual framework was proposed based on the literature [2, 3, 5–7], plausibility, the experience of the research team and the data available (Fig 1). First, we classified potential determinants in three sets of variables: characteristics of informal CG, CR characteristics and intensity of care. Then, we assumed that: 1. The three sets might affect the burden of care (CarerQoL-7D) as perceived by the CG; 2. The burden influences well-being/happiness (CarerQoL-VAS) and 3: The three sets can also influence well-being/happiness either directly or indirectly through their relationship with burden.

To end up with a parsimonious model we followed a stepwise approach: First, we fitted bivariate linear regressions for all clinically relevant variables to test for statistical significance

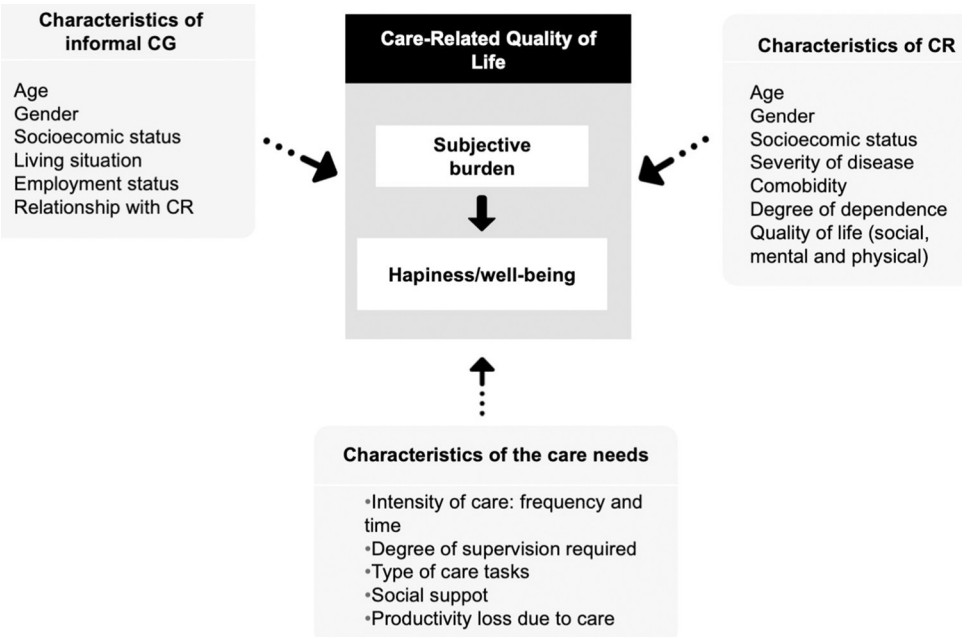

**Fig 1. Conceptual framework of components and determinants of care-related quality of life.**

of their association with the CarerQoL-7D utility value and the CarerQoL-VAS. Second, we performed three sets of multivariable models: Model 1 with CR characteristics, Model 2 with CG characteristics and Model 3 with the type and intensity of care. Finally, we kept all relevant variables found significant (using a liberal p value of 0.10) in the previous step and constructed structural equation models based on the proposed theoretical framework. Additionally, we constructed the equivalent model with a latent variable for the construct burden of care (instead of using utility weights). We used goodness of fit indices to test plausibility of the models: Chi squared test statistic, RMSEA (root mean square error of approximation); SRMR (Standardised Root Mean Square Residual); CFI (comparative fit index) [35].

Additionally, we evaluated internal consistency of the CarerQoL-7D in our sample by estimating Cronbach's Alpha.

## Ethical considerations

Approval was obtained from Hospital Vall d'Hebron Research Ethics Committee (reference PR(AG)229-2017).

## Results

Among the 629 patients included in the cost-utility study, 372 survived and were available at 6 months after stroke. From those, 132 had an informal CG, and both, the CG and the CR attended the telephone follow up. We were not able to assess the presence of an informal caregiver in 73 cases: 14 were lost to follow up or denied participation and 59 patients were unavailable at the time of follow up either because they were in a nursing home or hospitalized. A total of 132 caregiver/care receivers' pairs were included in this study. CG were predominantly female 98 (74.2%) with and average age of 59.4 ± 12.5 years, and 75 (56.8%) were partners of the CR. CRs were mainly men (58.3%), with a mean age of 71.67 ± 13.41 years. The sample characteristics, stratified by Barthel are shown in Table 1.

**Table 1. Informal caregiver and care receivers' characteristics stratified by care receivers' dependency.**

| Characteristics of informal caregivers | Total and severe (n = 29) | Moderate (n = 30) | Mild (n = 73) | Total (n = 132) | P-Value |
|---|---|---|---|---|---|
| Age, n years, m (SD) | 59.3 (12.5) | 61.7 (11.6) | 58.5 (12.8) | 59.4 (12.5) | 0.513 |
| Sex, female, n (%) | 22 (75.9) | 23 (76.7) | 53 (72.6) | 98 (74.2) | 0.889 |
| Highest educational level, n (%) | | | | | 0.897 |
| No studies or incomplete primary studies | 3 (10.7) | 2 (7.1) | 8 (11.0) | 13 (10.1) | |
| Complete primary studies | 9 (32.1) | 9 (32.1) | 28 (38.4) | 46 (35.7) | |
| Secondary studies or higher education | 16 (57.1) | 17 (60.7) | 37 (50.7) | 70 (54.3) | |
| Relationship with the care receiver, n (%) | | | | | 0.145 |
| Partner | 13 (44.8) | 17 (56.7) | 45 (61.6) | 75 (56.8) | |
| Son or daughter | 15 (51.7) | 8 (26.7) | 20 (27.4) | 43 (32.6) | |
| Father or mother | 0 (0.0) | 3 (10.0) | 4 (5.5) | 7 (5.3) | |
| Brother or sister | 0 (0.0) | 1 (3.3) | 4 (5.5) | 5 (3.8) | |
| Father or mother-in-law | 1 (3.4) | 1 (3.3) | 0 (0.0) | 2 (1.5) | |
| Living together with care receiver, n (%) | 22 (75.9) | 26 (86.7) | 63 (86.3) | 111 (84.1) | 0.390 |
| Living together with children or grandchildren, n (%) | 8 (27.6) | 7 (23.3) | 17 (23.3) | 32 (24.2) | 0.893 |
| **Characteristics of care receivers** | **Total and severe (n = 29)** | **Moderate (n = 30)** | **Mild (n = 73)** | **Total (n = 132)** | **P-Value** |
| Age, n years, m (SD) | 75.9 (11.6) | 73.5 (12.6) | 69.3 (14.0) | 71.67 (13.4) | 0.055 |
| Sex, female, n (%) | 17 (58.6) | 14/30 (46.7) | 24/73 (32.9) | 55/132 (41.7) | 0.048 |
| Pathological history* | | | | | |
| Hypertension | 22 (75.9) | 20 (66.7) | 48 (65.8) | 90 (68.2) | 0.601 |
| Dyslipidaemia | 14 (48.3) | 16 (53.3) | 37 (50.7) | 67 (50.8) | 0.927 |
| Current smoking | 3 (10.3) | 5 (16.7) | 16 (21.9) | 24 (18.2) | 0.381 |
| Enolism | 3 (10.3) | 2 (6.7) | 9 (12.3) | 14 (10.6) | 0.697 |
| Diabetes mellitus | 7 (24.1) | 9 (30.0) | 20 (27.4) | 36 (27.3) | 0.880 |
| Ischemic heart disease | 3 (10.3) | 4 (13.3) | 10 (13.7) | 17 (12.9) | 0.898 |
| Heart failure | 3 (10.3) | 1 (3.3) | 4 (5.5) | 8 (6.1) | 0.504 |
| Peripheral arterial disease | 2 (6.9) | 2 (6.7) | 3 (4.1) | 7 (5.3) | 0.793 |
| Previous TIA stroke | 4 (13.8) | 2 (6.7) | 16 (21.9) | 22 (16.7) | 0.151 |
| Atrial fibrillation | 9 (31.0) | 7 (23.3) | 14 (19.2) | 30 (22.7) | 0.434 |
| Previous anticoagulation | 6 (20.7) | 7 (23.3) | 11 (15.1) | 24 (18.2) | 0.567 |
| Baseline NIHSS, n (%) | | | | | 0.011 |
| Mild | 0 (0.0) | 0 (0.0) | 2 (2.9) | 2 (1.6) | |
| Moderate | 5 (17.9) | 8 (27.6) | 35 (50.0) | 48 (37.8) | |
| Severe or very severe | 23 (82.1) | 21 (72.4) | 33 (47.1) | 77 (60.6) | |
| Stroke etiology, n (%) | | | | | 0.865 |
| Ischemic** | 19 (65.5) | 22 (73.3) | 54 (74.0) | 95 (72.0) | |
| Hemorrhagic stroke | 9 (31.0) | 7 (23.3) | 16 (21.9) | 32 (24.2) | |
| Mimic stroke | 1 (3.4) | 1 (3.3) | 3 (4.1) | 5 (3.8) | |
| mRs 6 months, n (%) | | | | | < 0.001 |
| Mild | 0 | 0 | 5 | 5 (3.8) | |
| Moderate | 14 (48.3) | 28 (93.3) | 68 (93.2) | 110 (83.3) | |
| Severe | 15 (51.7) | 2 (6.7) | 0 | 17 (12.9) | |
| EQ-5D-5L index, m (SD) | 0.18 (0.29) | 0.41 (0.21) | 0.56 (0.25) | 0.44 (0.29) | < 0.001 |
| EQ-VAS, m (SD) | 34.46 (24.32) | 51.55 (22.4) | 54.73 (21.2) | 49.65 (23.41) | < 0.001 |

Abbreviations: m mean, SD standard deviation, n number. NIHSS, National Institute of Health Stroke Scale; mRs, modified Rankin Scale.

*Care receivers may have more than one disease or risk factor.

** Transient ischemic stroke (n = 2).

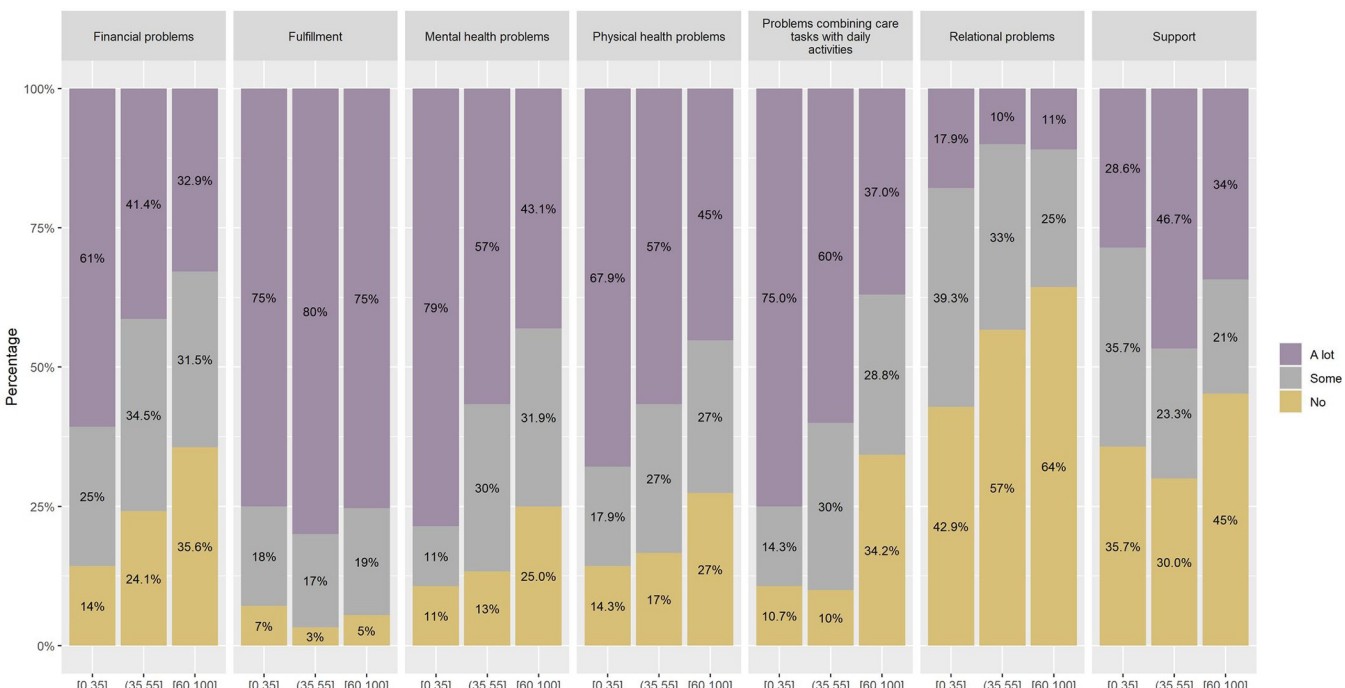

**Fig 2. Distribution of CarerQoL-7D domains reported by informal caregivers by degree of dependence of the care receivers.** Barthel index: [0,35] severe; [35,55] moderate; [60,100] mild.

The mean CarerQol-7D was significantly worse the higher the dependence: 45.45 (SD = 16.03) in severe dependence, 56.59 (SD = 20.89) in moderate and 57.05 (SD = 21.84)] in mild dependence. As shown in Fig 2, most CGs felt fulfilled with caregiving regardless of the degree of CR dependence, they had increasing financial problems, mental health problems, physical health problems, problems combining care tasks with daily activities and relational problems with increasing CR dependence. Mean CarerQol-VAS was significantly lower with increasing degree of dependence:4.73 (SD = 3.10) in severe, 5.50 (SD = 2.68) in moderate and 5.93(SD = 2.63) in mild dependence.

Table 2 describes the characteristics and intensity of care. The majority of CGs (86%) provided 7 days per week of care, and the number of hours dedicated per week ranged from a

**Table 2. Characteristics and intensity of care stratified by dependence of the care receivers.**

| | Total and severe (n = 29) | Moderate (n = 30) | Mild (n = 73) | Total (n = 132) | P-Value |
|---|---|---|---|---|---|
| Frequency of care is 7 days per week, n (%) | 25 (82.6) | 26 (89.7) | 61 (85.9) | 112 (86.8) | 0.876 |
| Total hours per week of care (m, SD; M, P25-P75) | 40.3 (19.8); 40 (21–57) | 38 (18.9); 35 (25–49) | 24 (16.8); 21 (10–37) | 30.7 (19.4);28 (16.5–41.5) | < 0.001 |
| Total hours of care per week, by type of activity, (m, SD; M, P25-P75) | | | | | |
| Household chores | 5.4 (7.0); 2 (0–10) | 5 (6.2); 3 (0–8) | 4.6 (6.7); 0 (0–7.5) | 4.9 (6.6); 1 (0–8) | 0.497 |
| Direct care | 29.6 (14.6); 30 (15–41) | 27.10 (17.1); 23.5 (14.7–36.2) | 13.4 (12.2); 10 (1.5–20.5) | 20.1 (15.7); 15.5 (7–30) | 0.794 |
| Support tasks | 5.2 (4.1); 5 (2–7) | 5.90 (3.9); 4.5 (2–10) | 5.97 (5.4); 5 (2–8.5) | 5.8 (4.8); 5 (2–8) | < 0.001 |
| The CR needs constant supervision | 20 (69.0) | 12 (40.0) | 6 (8.2) | 38 (28.8) | <0.001 |
| Care given by more than one person | 13 (44.8) | 6 (20.0) | 19 (26) | 38 (28.8) | 0.080 |

Values are expressed as number (%), mean ± standard deviation, median [interquartile range].

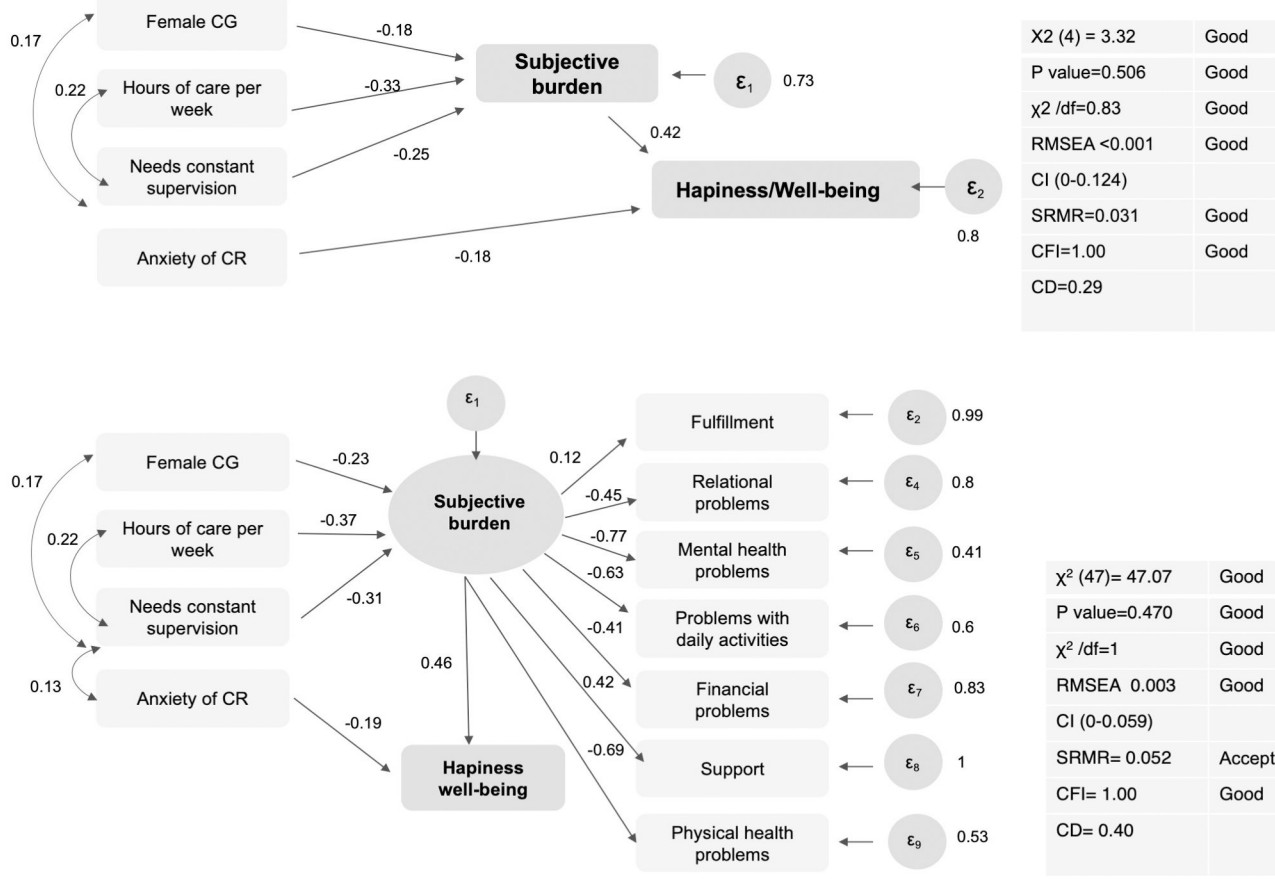

**Fig 3. Structural equation models showing the significant relationships between CG, CR and care characteristics with CG subjective burden and happiness/well-being. A:** using utility weights of the CarerQoL-7D; **B:** using a latent variable for burden as a function of the 7 dimensions of the CarerQoL-7D. Variables in a square are observed variables. Variables in a circle represent a latent construct. Small circles are the error terms. Arrows with one arrowhead represent associations and standardized coefficients; arrows with two arowheads represent covariances. RMSEA: root mean square error of approximation; CI: Confidence Interval; SRMR: Standardised Root Mean Square Residual; CFI: Comparative Fit Index; CD: Coefficient of Determination.

median of 21 for mild dependency to 40 for total/severe dependency. More than half of this time was spent on direct care.

In bivariate analysis (S1 Table), CG characteristics such as being female and being a brother or sister of the CR were associated with CarerQoL-7D. No significant associations were found between CG characteristics and happiness. More severe stroke, higher dependency, and worse functional status of the CR were significantly associated with lower scores of CarerQol-7D and CarerQol-VAS. CR quality of life was correlated with informal CG quality of life.

Fig 3 shows significant associations found in structural equation models and goodness-of-fit statistics. Fig 3A shows the structural equation model analysis using utility weights for caregiver perceived burden, and Fig 3B shows the same model using a latent variable for burden as a function of the 7 dimensions of the CarerQoL-7D. Both models showed significant associations between being a woman, more hours of care per week, and CR need for constant supervision with higher burden. Higher burden and anxiety/depression of the CR were associated with lower levels of happiness/well-being of the CG. Burden estimated as a latent variable was mainly explained by the negative factors, having problems with own mental and physical health, having problems to combine daily activities with caring tasks, having problems in the relationship with

the CR, and having economic problems. Goodness-of-fit values were within the recommended rules for model evaluation [35]. Cronbach's Alpha value for the CarerQoL-7D was 0.63, indicating low correlation between items in general, but especially for Fulfillment and Support from others, (all values for item-test and item-rest correlations can be found in S2 Table.

## Discussion

We describe caregiving characteristics and its consequences on the informal CG stratified by the degree of functional dependence of the CR in terms of burden and happiness/well-being. The majority of CGs were women (mostly spouses) and they experienced significantly higher burden than men, independently of the degree of dependency of the CR. Intensity of care and the burden perceived by the CG was substantial and increased with the degree of functional dependence of the CR. Female gender and intensity of care were the main determinants of subjective burden. While subjective burden and anxiety/depression of the CR were correlated with lower levels of CG happiness/wellbeing.

Despite an increase in all problems related to caregiving with increasing levels of dependence, almost all caregivers felt fulfilled with the caregiving task [4].

Our results show an association between increased caregiver burden and time spent caring, in line with previous research [14]. CG burden also increased when the CRs needed constant vigilance, regardless of the degree of dependency. The intensity of caregiving had a negative impact on CG perceived burden. There is variability in the literature regarding the number of hours of informal care for stroke survivors [2, 36]. In our study, CGs provided an average of 30.7 hours of care per week, which corresponds to 76.8% of a usual working week of 40 hours of paid work. The highest number of informal care hours was reported for direct care, followed by support and household chores [2].

The fact that caregiving is mainly assumed by women (mostly spouses, but also daughters) is consistent with previous studies [2]. Gender differences in informal caregiving may be related to the existing culture, where women internalize the role of caregiving [37] and are more willing than men to leave work to be able to take on caregiving tasks. Not only are mostly women who assume caregiving more often, but they also experience the impact of caregiving differently. Women's greater perception of health deterioration may be due to the unequal distribution of caregiving and daily living responsibilities [38].

In contrast to previous studies [14] we did not found a correlation between the level of education and perceived burden. This difference might be explained by the use of different outcome measures in other studies. We also observed an association between patients' level of anxiety or depression and CG happiness/wellbeing [4].

The present study has some limitations and strengths. Firstly, as a secondary analysis of data collected in the RACECAT clinical trial, it is not representative of the range of stroke severity but is biased towards those with more severe strokes at onset, thus accounting for a population with higher chances of disabling sequelae and of caring needs. For the same reason, we did not measure some aspects that might influence caregiving burden, such as cognitive status of the CR or socioeconomic level; and sample size was not calculated based on the objectives of this study and some subgroups might be underrepresented, thus limiting the interpretation of some of the results. Secondly, the associations that emerged cannot be interpreted as casual paths, as many relationships might be bidirectional. Finally, as some data was collected using self-reported questions, could imply recall and/or comprehension bias. For instance, hours of caring activities could be either underestimated by the caregiver due to the difficulty to recall or to discern the boundaries of informal care, or overestimated due to social desirability or the expectation of obtaining external help.

Data collection within a clinical trial promoted completeness and quality of the data. An additional strength is the stratification by Barthel index, an indicator used in most health and social care systems, which will allow our estimates to be extrapolated to other contexts and to foresee the impact of new policies or interventions. Furthermore, the use of CarerQol-7D instrument is an advantage compared to previous studies, as it allows the measurement of positive dimensions of care. The fact that internal consistency of the CarerQoL-7D is low indicates that each item is measuring different aspects of care, and that they are not necessarily correlated with each other. This is especially relevant when we intend to measure the positive aspects of care, such as satisfaction or fulfillment, which might be independent from financial or other practical problems. However, it also points out that the overall score of the CarerQoL-7D should be interpreted with caution and taking into account the values for the individual dimensions.

Our findings suggest the importance of strengthening interventions for early detection of caregiver burden and implementing policy improvements to support CGs of stroke survivors. Support measures should be multi-faceted and cross-sectoral between government departments. The support programs should focus on the mental and emotional health of the stroke survivor and CG [39]. Furthermore, it highlights the importance of access to respite care for CG's stroke survivors and the need for improvements in work-life balance policies. Considering the heterogeneity among CG, it should incorporate the gender perspective to tailor recourses to the differential needs of men and women [37, 38].

There are opportunities for future research. First, specific studies addressing gender differences in informal caregiving and its differential impact on health are needed. Secondly, the positive experiences reported may have acted as a buffer against negative caregiving outcomes. Therefore, understanding of the factors influencing some caregivers, who appear to maintain well-being and health, would be of help to develop person-centered interventions. Finally, we need to develop Spanish of utility weights for the CarerQol-7D.

## Conclusions

This study presents the consequences of caregiving intensity according to degree of dependence and the identification of possible factors associated with caregiving-related quality of life of informal CG after stroke. The findings suggest the importance of developing two types of support interventions for CG: respite and psychosocial support. Especially for women with high caring burden and/or caring for persons with high levels of anxiety or depression.

## Supporting information

**S1 Appendix. STROBE statement.**
(DOCX)

**S1 Table. Bivariate analysis, CarerQol-7D/CarerQol-VAS with characteristics of caregivers and care receivers.**
(DOCX)

**S2 Table. Internal consistency of the Carer-QoL-7D.**
(DOCX)

## Acknowledgments

The authors wish to acknowledge the steering committee and the investigators of the RACE-CAT trial for providing part of the data and access to the study participants. Statistical analysis

has been partially carried out in the Statistics and Bioinformatics Unit (UEB) Vall d'Hebron Hospital Research Institute (VHIR).

## Author Contributions

**Conceptualization:** Lorena Villa-García, John Slof, Sònia Abilleira, Aida Ribera.

**Data curation:** Lorena Villa-García, Mercè Salvat-Plana, John Slof, Natalia Pérez de la Ossa, Sònia Abilleira, Marc Ribó, Verónica Hidalgo-Benítez, Aida Ribera.

**Formal analysis:** Lorena Villa-García, John Slof, Aida Ribera.

**Funding acquisition:** Aida Ribera.

**Supervision:** Aida Ribera.

**Visualization:** Lorena Villa-García.

**Writing – original draft:** Lorena Villa-García, Aida Ribera.

**Writing – review & editing:** Mercè Salvat-Plana, John Slof, Natalia Pérez de la Ossa, Sònia Abilleira, Marc Ribó, Verónica Hidalgo-Benítez, Marco Inzitari, Aida Ribera.

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
