## [Decision Letter · Decision Letter 0]

4 Mar 2024

PONE-D-23-42917Care-related Quality of Life of informal caregivers of stroke survivorsPLOS ONE

Dear Dr. Ribera,

Thank you for submitting your manuscript to PLOS ONE. After careful consideration, we feel that it has merit but does not fully meet PLOS ONE’s publication criteria as it currently stands. Therefore, we invite you to submit a revised version of the manuscript that addresses the points raised during the review process.

We look forward to receiving your revised manuscript.

Kind regards,

Ahmed Mohamed Elhfnawy

Academic Editor

PLOS ONE

Journal Requirements:

Whilst you may use any professional scientific editing service of your choice, PLOS has partnered with both American Journal Experts (AJE) and Editage to provide discounted services to PLOS authors. Both organizations have experience helping authors meet PLOS guidelines and can provide language editing, translation, manuscript formatting, and figure formatting to ensure your manuscript meets our submission guidelines. To take advantage of our partnership with AJE, visit the AJE website (http://aje.com/go/plos) for a 15% discount off AJE services. To take advantage of our partnership with Editage, visit the Editage website (www.editage.com) and enter referral code PLOSEDIT for a 15% discount off Editage services. If the PLOS editorial team finds any language issues in text that either AJE or Editage has edited, the service provider will re-edit the text for free.

This study received funding by the Fundació Marató de TV3 (ref. 19/U/2017; authors A.R who received.   

LV-G was funded by the Industrial Doctorates Program [reference 2020 DI 76], promoted by the Government of Catalonia, Spain. 

Reviewers' comments:

Reviewer's Responses to Questions

**Comments to the Author**

1. Is the manuscript technically sound, and do the data support the conclusions?

Reviewer #1: Yes

Reviewer #2: Yes

Reviewer #3: No

2. Has the statistical analysis been performed appropriately and rigorously? 

Reviewer #1: Yes

Reviewer #2: Yes

Reviewer #3: No

3. Have the authors made all data underlying the findings in their manuscript fully available?

Reviewer #1: Yes

Reviewer #2: No

Reviewer #3: No

4. Is the manuscript presented in an intelligible fashion and written in standard English?

Reviewer #1: Yes

Reviewer #2: Yes

Reviewer #3: No

5. Review Comments to the Author

Reviewer #1: Thank you for inviting me to review this paper on qol in caregivers of stroke survivors. This is an important and timely topic. I have a few minor comments that can help strengthen the submission:

Minor:

-In Abstract, details regarding the statistical analysis in methods is lacking - what factors were examined as predictors of QoL? Were there a priori hypotheses or was this exploratory? This does not become clear until later in the manuscript

—Cannot fully understand the models based on abstract methods; would specify that you examined 2 dependent variables (QoL and happiness)

-Throughout the text the authors refer to bivariate analysis (unadjusted regressions) as univariate analysis. Please change this. Univariate statistics refers to descriptive statistics.

Intro:

-the first two sentences in paragraph 3 are not necessarily linked. Sentence 1 states that CG burden is the product of complex interaction of factors. You can connect sentence 2 by starting with “As a result of the complex interaction of these factors…”

-lines 70-72 - I’m not sure policy decisions would be based on solely economic evaluations and thus this sentence is not necessary. Would consider substituting this sentence out…

Stats:

-For the conceptual model, I would also add dyadic relationship quality as a predictor of cg QOL. I recognize that this was secondary data analysis of an RCT, and you properly don't have this measure. Thus, I would consider mentioning this missing variable in limitations.

-what was the significance value you examined for bivariate linear regressions? Please report. Also, consider using a liberal value of 0.10 - this can help include variables that would have synergistic effects with other variables when included in the same multivariable model.

-Can you discuss further how hours per week was measured/reported? Was this self-reported by the cg? Do we have a sense of how rigorous the measure is (i.e,. Would cgs estimate the amount of time caregiving or was there a more standardized way to measure this)?

—If self-report, I am wondering if this could have been biased based on # of hours cg need to report caregiving for other benefits (e.g,. Social desirability, government benefits (if any), being help-seeking, etc…)

Results:

Can you clarify - was it 59 caregivers or 59 patients that were unavailable bc of being in nursing home/hospitalized?

General:

There is a ton of data presented here, and I recommend the authors tighten the presentation and streamline to highlight the most important results.

Reviewer #2: A nicely written article and well designed study, I have only one comment regarding the instrument validation. I recommend minor revision.

Title - indicate study type

Abstract

- No data in results, insert number of participants, and other mentioned data

Introduction

- Sound and relevant

- Define quality of life, measures of qol - general and specific

- Are there previous studies similar to yours?

Methods

- Indicate study type

- How was the CarerQoL-7D obtained? Did you use a validated translation or did you translate it on your own? Is the instrument free or do you need a licence?

- Why and how did you categorized BI into 3 categories? ( We categorized BI into severe (<35),moderate (40-55) and mild (>60) dependence)? What if someone had 38 points or 57?

Results

As this is the first time that the scale is used on stroke patients at least Cronbach alpha of the subscales has to be calculated and presented in the results.

Discussion

Sound and pertinent.

Reviewer #3: I have reviewed the manuscript and identified several areas for improvement:

The topic of stroke among the population is not novel, but the specific research gap is not clearly articulated in the study's title. The abstract lacks sufficient detail, particularly regarding the objectives and methodology. It should include criteria for selecting caregivers and stroke patients, the type of assessment tool used, and the scoring system employed. Additionally, the study design—whether qualitative or quantitative—should be explicitly stated, along with justification for the need to conduct interviews. The findings should be presented with clarity regarding their intensity and direction, ensuring alignment with the study's conclusion as outlined in the abstract.

The introduction lacks a solid theoretical framework to support the arguments, and the research gaps are not adequately explained, given the prevalence of similar studies. It would be beneficial to contextualize the research within the current changes in the local healthcare system and its evolving needs. While it's understood that this work is part of a larger economic evaluation study, the manuscript lacks detail and clarity, particularly regarding the development of the assessment tool and the rationale for conducting interviews.

The methodology section requires greater clarity, with detailed explanations of both dependent and independent variables. Potential confounding factors, such as stroke severity, duration, and caregivers' education levels, should be identified and controlled for. The data collection process, sample selection criteria, and sample size determination should be clearly outlined. Although previous studies have identified factors contributing to caregivers' poor quality of life, it's unclear why a structural equation model analysis was chosen over logistic or linear regression analysis. Justification for this choice should be provided, along with a conceptual framework integrated with the theoretical framework in the introduction.

The results should align closely with the objectives and methodology of the study. Any mention of follow-up procedures, as indicated in Figure 2, should be clearly explained, along with the final number of participants included in the study. Consideration should be given to the presentation format of Figure 2, as it may not be essential for this manuscript if it does not pertain to a cohort design. Additionally, converting the information in Figure 3 into a table format may enhance clarity. Table 2 appears to serve dual purposes by describing both the intensity of care and providing comparative analysis, which could be confusing without clear delineation of objectives.

Given the unequal gender distribution in the sample, particularly with a significant representation of females, caution should be exercised when discussing gender-related findings. The limitations outlined in the study should be acknowledged, as they reveal weaknesses in the scope of the manuscript. These limitations should be considered when interpreting the results and discussing their implications.

6. PLOS authors have the option to publish the peer review history of their article (what does this mean?). If published, this will include your full peer review and any attached files.

Reviewer #1: No

Reviewer #2: **Yes: **Ksenija Bazdaric

Reviewer #3: No

---

## [Author Response · Author response to Decision Letter 0]

5 Apr 2024

Reviewer #1: Thank you for inviting me to review this paper on qol in caregivers of stroke survivors. This is an important and timely topic. I have a few minor comments that can help strengthen the submission:

Comment 1

Minor:

-In Abstract, details regarding the statistical analysis in methods is lacking - what factors were examined as predictors of QoL? Were there a priori hypotheses or was this exploratory? This does not become clear until later in the manuscript

Answer 1

We wish to thank the reviewer for the general positive comment

We agree that the methods need more clarification in the Abstract. Thank you for pointing this out. We have added more information about the analysis to the abstract section. We believe that the new wording will help the reader to understand the general structure of the model and the a priori hypotheses. 

Changes in the text: 

Page 2 lines 41-43: “We evaluated the association between characteristics of informal caregiver, characteristics of care receiver and intensity of care, and the caregiver’s care-related quality of life (subjective burden and happiness) in a hypothesized model using structural equation modeling.”

Comment 2

—Cannot fully understand the models based on abstract methods; would specify that you examined 2 dependent variables (QoL and happiness)

Answer 2

As stated in the answer to comment 1, we examined the effect on the two components of quality of life: subjective burden and happiness. We hope that this is clearer now

Comment 3

-Throughout the text the authors refer to bivariate analysis (unadjusted regressions) as univariate analysis. Please change this. Univariate statistics refers to descriptive statistics.

Answer 3

As suggested by the reviewer, we have changed the wording throughout the text. 

Comment 4

Intro:

-the first two sentences in paragraph 3 are not necessarily linked. Sentence 1 states that CG burden is the product of complex interaction of factors. You can connect sentence 2 by starting with “As a result of the complex interaction of these factors…”

Answer 4

We have changed the sentence as suggested. Page 3, lines 70-71. 

“As a result of the complex interaction of these factors, CGs may experience impaired mental or physical health, and reduced quality of life3,5. Furthermore, their participation in leisure activities, social relationships, and/or paid work may be limited due to caregiving responsibilities” 

Comment 5

-lines 70-72 - I’m not sure policy decisions would be based on solely economic evaluations and thus this sentence is not necessary. Would consider substituting this sentence out…

Answer 5

As suggested, we have deleted this sentence. We agree that the statement might be too speculative

Comment 6

Stats:

-For the conceptual model, I would also add dyadic relationship quality as a predictor of cg QOL. I recognize that this was secondary data analysis of an RCT, and you properly don't have this measure. Thus, I would consider mentioning this missing variable in limitations.

Answer 6

We totally agree with the reviewer. However, an item regarding “relational problems” is already included in the CarerQoL instrument, thus we could not use it as a predictor. What the model shows is that this item has a relatively high weight on subjective wellbeing (coefficient=-0.45) and this is highlighted in pag 11 line 239-241. 

We also evaluated the bivariate relationship between Relational problems and happiness (CarerQoL-VAS) and we did not find a significant association (supplementary table S3). This might be because the majority of caregivers reported they did not have relational problems with care receivers (as shown in Figure 3). 

Comment 7

-what was the significance value you examined for bivariate linear regressions? Please report. Also, consider using a liberal value of 0.10 - this can help include variables that would have synergistic effects with other variables when included in the same multivariable model.

Answer 7

We used a p value of 0.05 as significance level to select the variables for the final SEM models. We have checked again our analyses and we can confirm that the selection would have been the same considering the more liberal value of 0.10. Therefore, we have added this significance level in the methods section. 

Changes in text: page 7, lines 176-177: “(using a liberal p value of 0.10)”

Comment 8

-Can you discuss further how hours per week was measured/reported? Was this self-reported by the cg? Do we have a sense of how rigorous the measure is (i.e,. Would cgs estimate the amount of time caregiving or was there a more standardized way to measure this)?

—If self-report, I am wondering if this could have been biased based on # of hours cg need to report caregiving for other benefits (e.g. Social desirability, government benefits (if any), being help-seeking, etc…)

Answer 8

Thank you for this suggestion. Hours of caregiving were assessed with rigorous and standardized telephone interview (Hoefman RJ, Van Exel NJA, Brouwer WBF. iMTA Valuation of Informal Care Questionnaire (iVICQ). Version 1.0 (December 2011). Rotterdam: iBMG / iMTA, 2011. [retrieved from www.bmg.eur.nl/english/imta/publications/manuals_questionnaires/ on dd/mm/yyyy]

) performed by a trained nurse in the context of the RACECAT clinical trial. However, we agree that, it might be difficult for the caregivers to recall the number of caring hours and even to discern the boundaries of informal care (i.e to separate hours of direct care from hours of companionship or living together). 

Changes in text

Page 6, line 145. We have added the reference for the instrument to evaluate care hours:” following a standardized questionnaire adapted from de iVICQ (cita). “

This has been added as a limitation in page 12, lines 281-284: “Finally, as some data was collected using self-reported questions, could imply recall and/or comprehension bias. For instance, hours of caring activities could be either underestimated by the caregiver due to the difficulty to recall or to discern the boundaries of informal care, or overestimated due to social desirability or the expectation of obtaining external help.”

Comment 9

Results:

Can you clarify - was it 59 caregivers or 59 patients that were unavailable bc of being in nursing home/hospitalized?

Answer 9

Thank you for pointing this out. The numbers refer to patients lost to follow up. We have clarified this in page 8, line 185. 

Comment 10

General:

There is a ton of data presented here, and I recommend the authors tighten the presentation and streamline to highlight the most important results.

Answer 10

We totally agree with the reviewer. We consider this work as exploratory, highlighting several points that deserve further attention in future studies, like the gender perspective and the need to focus on the mental and emotional health of patients to improve wellbeing of caregivers. We have made the effort to keep at the minimum the relevant information presented in the main text, but preserving other results in the supplementary material. 

Reviewer #2: A nicely written article and well designed study, I have only one comment regarding the instrument validation. I recommend minor revision.

Comment 1

Title - indicate study type

Answer 1

We thank the reviewer for the general positive comment. 

We have added the study type in the title, as suggested: “Care-related Quality of Life of informal caregivers of stroke survivors: cross-sectional analysis of a randomized clinical trial”. 

Comment 2

Abstract

- No data in results, insert number of participants, and other mentioned data

Answer 2

We agree with the reviewer’s assessment. Accordingly, we have added numeric data (numbers of participants and coefficients of the associations) in the abstract. 

Changes in text: Pag 2, Line 44-45. “Of the 132 caregivers, 74,2% were women with an average age of 59.4 ± 12.5 years. The 56.8% of them were spouses. The care intensity ranged from a mean of 24h/week for mild to 40h/week for severe dependence. Most caregivers (76.3%) were satisfied with their task, regardless of dependence, but showed increasing problems in caring for severely dependent persons. Being a woman (coeff. -0.23; 95%CI: -0.40, -0.07), spending more time in care tasks (coeff -0.37; -0.53, -0.21) and care receiver need of constant supervision (coeff 0.31; -0.47, -0.14) were associated with higher burden of care, irrespective of the degree of dependence. Caregiver burden (coeff 0.46; 0.30-0.61) and care receiver anxiety or depression (coeff -0.19; -0.34, -0.03) were associated with lower caregiver happiness.”

Comment 3

Introduction

- Sound and relevant

- Define quality of life, measures of qol - general and specific

Answer 3

We have added in introduction Pag 3, line 81-86 “The WHO Health Organization defines QOL as an individual's perception of his or her own life in the context of the culture and value systems in which they live and in relation to their expectations, norms, and concerns (ref 17). Measures of QoL can be generic (i.e., designed to be used across disorders and health states) or disease/affliction-specific (i.e., related to a single disorder or health state). Several methods have been proposed to determine an appropriate definition and measurement of QoL in the context of informal care, however, as yet no unified agreement has been established” 

Comment 4

- Are there previous studies similar to yours? 

Answer 4

We have added in page 4, lines 90-92 “However, to our knowledge there are no studies evaluating the CG-related quality of life of CG of patients with stoke sequelae in terms of burden and well-being as assessed by the CarerQoL”

Comment 5

Methods

- Indicate study type

Answer 5

Thank you for this suggestion. We have added the study type in the methods. Page 4, Line 102 “Cross-sectional analysis of prospective data collected in a cost-utility study alongside the RACECAT trial…”

Comment 6

- How was the CarerQoL-7D obtained? Did you use a validated translation or did you translate it on your own? Is the instrument free or do you need a licence?

Answer 6

We used the Spanish version provided by the iMTA (Institute for Medical Technology Assessment). The CarerQoL part of the iMTA valuation of informal care, which provides all the instruments to perform economic evaluation of care in different languages: hours of informal care, hours of productivity loss and the CarerQoL. The CarerQoL is available for use without prior permission from the authors. 

We have added the clarification in the manuscript. Page 5, Line 122: “The main outcome was caregiving-related quality of life measured with the Spanish version of the CarerQoL” 

Comment 7

- Why and how did you categorized BI into 3 categories? ( We categorized BI into severe (<35),moderate (40-55) and mild (>60) dependence)? What if someone had 38 points or 57?

Answer 7

BI is usually categorized in into five groups: independent, mild, moderate, severe and total dependence; we grouped independent with mild dependence, and severe with total dependence because of the very small sample size in the extreme values .The BI score is not rigorously continuous, as it has no values between 35 and 40, neither between 55 and 60.

Comment 8

Results

As this is the first time that the scale is used on stroke patients at least Cronbach alpha of the subscales has to be calculated and presented in the results.

Answer 8

We measured internal consistency using our data and found a Cronbach alpha value of 0.63, which is in line with previous articles (not performed with carers of stroke patients). As other authors have already stated, internal consistency of the CarerQoL is low, but it has to be noted that this is not a unidimensional instrument. ON the contrary, each item of the CarerQoL is measuring a different aspect of care, and different aspects of care are not necessary correlated with each other. Although we agree with the reviewer that this is a relevant issue, assessing the psychometric properties of the CarerQoL is not within the objectives of our study, so that, for the sake of simplicity, we prefer not to include this information. However, we can consider including this analysis if requested.

Discussion

Sound and pertinent.

Reviewer #3: I have reviewed the manuscript and identified several areas for improvement:

Comment 1

The topic of stroke among the population is not novel, but the specific research gap is not clearly articulated in the study's title. The abstract lacks sufficient detail, particularly regarding the objectives and methodology. It should include criteria for selecting caregivers and stroke patients, the type of assessment tool used, and the scoring system employed. Additionally, the study design—whether qualitative or quantitative—should be explicitly stated, along with justification for the need to conduct interviews. The findings should be presented with clarity regarding their intensity and direction, ensuring alignment with the study's conclusion as outlined in the abstract.

Answer 1

We believe that our study, even if not novel, adds more evidence on an important topic, so there is not a research gap that can be clearly stated in the study title. However, we have changed the title to clarify the study design: “Care-related Quality of Life of informal caregivers of stroke survivors: cross-sectional analysis of a randomized clinical trial”.

We agree that the methods need more clarification in the Abstract. Thank you for pointing this out. We have added more information about the analysis to the abstract section. We believe that the new wording will help the reader to understand the methodological approach and the general structure of the model and the a priori hypotheses we tested. 

Changes in the text: 

Page 2 lines 41-43: “We evaluated the association between characteristics of informal caregiver, characteristics of care receiver and intensity of care, and the caregiver’s care-related quality of life (subjective burden and happiness) in a hypothesized model using structural equation modeling.”

Also, we have added numeric data (numbers of participants and coefficients of the associations) in the abstract. 

Changes in text: Pag 2, Line 44-45. “Of the 132 caregivers, 74,2% were women with an average age of 59.4 ± 12.5 years. The 56.8% of them were spouses. The care intensity ranged from a mean of 24h/week for mild to 40h/week for severe dependence. Most caregivers (76.3%) were satisfied with their task, regardless of dependence, but showed increasing problems in caring for severely dependent persons. Being a woman (coeff. -0.23; 95%CI: -0.40, -0.07), spending more time in care tasks (coeff -0.37; -0.53, -0.21) and care receiver need of constant supervision (coeff 0.31; -0.47, -0.14) were associated with higher burden of care, irrespective of the degree of dependence. Caregiver burden (coeff 0.46; 0.30-0.61) and care receiver anxiety or depression (coeff -0.19; -0.34, -0.03) were associated with lower caregiver happiness”

Comment 2

The introduction lacks a solid theoretical framework to support the arguments, and the research gaps are not adequately explained, given the prevalence of similar studies. It would be beneficial to contextualize the research within the current changes in the local healthcare system and its evolving needs. While it's understood that this work is part of a larger economic evaluation study, the manuscript lacks detail and clarity, particularly regarding the development of the assessment tool and the rationale for conducting interviews.

Answer 2

Following the reviewer suggestions we have added some sentences in the introduction, specifically: the definition of quality of life and the specificities of the CarerQoL. 

Page 3, lines 80-82: “The WHO Health Organization defines QOL as an individual's perception of his or her own life in the context of the culture and value systems in which they live and in relation to their expectations, norms, and concerns 

Evolving needs of current healthcare systems is stated in page 4, lines 93-96: “The future availability of informal care is of great social concern 21. Given the current transformations of health and social care integration22 and the increasing importance of the role of CGs, it is essential to le

---

## [Decision Letter · Decision Letter 1]

9 Jul 2024

PONE-D-23-42917R1Care-related Quality of Life of informal caregivers of stroke survivors: cross-sectional analysis of a randomized clinical trialPLOS ONE

Dear Dr. Ribera,

Thank you for submitting your manuscript to PLOS ONE. After careful consideration, we feel that it has merit but does not fully meet PLOS ONE’s publication criteria as it currently stands. Therefore, we invite you to submit a revised version of the manuscript that addresses the points raised during the review process.

**ACADEMIC EDITOR: **

I am looking forward to accept the revised manuscript with the condition that the comments by Reviewer 2 regarding reporting the internal consistency reliability of the tool, as well as further elaborations on restrictions for data availability. Please refer to PLOS Data Policy (https://journals.plos.org/plosone/s/data-availability). 

If there are ethical or legal restrictions on sharing a sensitive data set, authors should provide the following information within their Data Availability Statement upon submission:

Explain the restrictions in detail (e.g., data contain potentially identifying or sensitive patient information)Provide contact information for a data access committee, ethics committee, or other institutional body to which data requests may be sent

We look forward to receiving your revised manuscript.

Kind regards,

Chai-Eng Tan

Academic Editor

PLOS ONE

Journal Requirements:

Reviewers' comments:

Reviewer's Responses to Questions

**Comments to the Author**

1. If the authors have adequately addressed your comments raised in a previous round of review and you feel that this manuscript is now acceptable for publication, you may indicate that here to bypass the “Comments to the Author” section, enter your conflict of interest statement in the “Confidential to Editor” section, and submit your "Accept" recommendation.

Reviewer #2: All comments have been addressed

Reviewer #3: All comments have been addressed

2. Is the manuscript technically sound, and do the data support the conclusions?

Reviewer #2: Yes

Reviewer #3: Yes

3. Has the statistical analysis been performed appropriately and rigorously? 

Reviewer #2: (No Response)

Reviewer #3: Yes

4. Have the authors made all data underlying the findings in their manuscript fully available?

Reviewer #2: No

Reviewer #3: No

5. Is the manuscript presented in an intelligible fashion and written in standard English?

Reviewer #2: Yes

Reviewer #3: Yes

6. Review Comments to the Author

Reviewer #2: Dear author, I am satisfied with the revision but please do report the internal consistency of 0.63 for the sake of transparency and share your data.

Reviewer #3: I have read the revised version of the manuscript and am satisfied with the corrections made. Thank you for taking into consideration all the comments and suggestions. There are numerous improvements, making the manuscript much better and easier to read. Well done.

7. PLOS authors have the option to publish the peer review history of their article (what does this mean?). If published, this will include your full peer review and any attached files.

Reviewer #2: No

Reviewer #3: No

---

## [Author Response · Author response to Decision Letter 1]

13 Jul 2024

Dear Editors and Reviewers of the Plos ONE, 

Thank you for the opportunity of resubmitting our work

Hereby we include a one by one answer to the editor and the reviewers’ comments. 

1. Regarding data sharing, data for the present analysis was obtained through interviews with stroke patients and their families after informed consent. The informed consent as approved by the Hospital Vall d’Hebron Research Ethics Committee did not include a specific consent for data sharing, thus data cannot be publicly available for ethical reasons.

2. Following the suggestion of reviewer #2 we have added the information related to internal consistency of the CarerQoL-7D

In the methods section, page 7, lines 178-179: “Additionally, we evaluated internal consistency of the CarerQoL-7D in our sample by estimating Cronbach’s Alpha”

In the results section, page 11, lines 241-243: “Cronbach’s Alpha value for the CarerQoL-7D was 0.63, indicating low correlation between items in general, but especially for Fulfillment and Support from others, (all values for item-test and item-rest correlations can be found in supplemental Table S2”

In the discussion section, page 14, lines 297-302: “The fact that internal consistency of the CarerQoL-7D is low indicates that each item is measuring different aspects of care, and that they are not necessarily correlated with each other. This is especially relevant when we intend to measure the positive aspects of care, such as satisfaction or fulfillment, which might be independent from financial or other practical problems. However, it also points out that the overall score of the CarerQoL-7D should be interpreted with caution and taking into account the values for the individual dimensions. “

We have added Table S2 in the Supplemental material: 

Table S2. Internal consistency of the Carer-QoL-7D

 Alpha Item-test correlation Item-rest correlation*

Fulfillment 0.64 0.30 0.12

Relational problems 0.59 0.53 0.34

Mental health problems 0.51 0.72 0.55

Financial problems 0.53 0.68 0.48

Problems with daily activities 0.60 0.56 0.33

Support from others 0.68 0.35 0.07

Physical health problems 0.63 0.70 0.52

Total scale 0.63 

Alpha: Cronbach’s Alpha values for the scales composed by all items but the one indicated in each row

Item-test correlation: correlation between the item and the total scale

Item-rest correlation: correlation between the item and the scale composed by the rest of items

---

## [Editor Report · Decision Letter 2]

16 Jul 2024

Care-related Quality of Life of informal caregivers of stroke survivors: cross-sectional analysis of a randomized clinical trial

PONE-D-23-42917R2

Dear Dr. Ribera,

We’re pleased to inform you that your manuscript has been judged scientifically suitable for publication and will be formally accepted for publication once it meets all outstanding technical requirements.

Kind regards,

Chai-Eng Tan

Academic Editor

PLOS ONE
---

## [Editor Report · Acceptance letter]

1 Aug 2024

PONE-D-23-42917R2 

PLOS ONE

Dear Dr. Ribera, 

I'm pleased to inform you that your manuscript has been deemed suitable for publication in PLOS ONE. Congratulations! Your manuscript is now being handed over to our production team.

Kind regards, 

on behalf of

Dr. Chai-Eng Tan 

Academic Editor

PLOS ONE